# Comprehensive Review on the Use of Artificial Intelligence in Ophthalmology and Future Research Directions

**DOI:** 10.3390/diagnostics13010100

**Published:** 2022-12-29

**Authors:** Nicoleta Anton, Bogdan Doroftei, Silvia Curteanu, Lisa Catãlin, Ovidiu-Dumitru Ilie, Filip Târcoveanu, Camelia Margareta Bogdănici

**Affiliations:** 1Faculty of Medicine, University of Medicine and Pharmacy “Grigore T. Popa”, University Street, No 16, 700115 Iasi, Romania; 2Department of Chemical Engineering, Cristofor Simionescu Faculty of Chemical Engineering and Environmental Protection, Gheorghe Asachi Technical University, Prof.dr.doc Dimitrie Mangeron Avenue, No 67, 700050 Iasi, Romania; 3Department of Biology, Faculty of Biology, “Alexandru Ioan Cuza” University, Carol I Avenue, No 20A, 700505 Iasi, Romania

**Keywords:** artificial intelligence, artificial intelligence in medicine, neural networks in ophthalmology, applications of artificial intelligence in ophthalmology, glaucoma and artificial intelligence

## Abstract

Background: Having several applications in medicine, and in ophthalmology in particular, artificial intelligence (AI) tools have been used to detect visual function deficits, thus playing a key role in diagnosing eye diseases and in predicting the evolution of these common and disabling diseases. AI tools, i.e., artificial neural networks (ANNs), are progressively involved in detecting and customized control of ophthalmic diseases. The studies that refer to the efficiency of AI in medicine and especially in ophthalmology were analyzed in this review. Materials and Methods: We conducted a comprehensive review in order to collect all accounts published between 2015 and 2022 that refer to these applications of AI in medicine and especially in ophthalmology. Neural networks have a major role in establishing the demand to initiate preliminary anti-glaucoma therapy to stop the advance of the disease. Results: Different surveys in the literature review show the remarkable benefit of these AI tools in ophthalmology in evaluating the visual field, optic nerve, and retinal nerve fiber layer, thus ensuring a higher precision in detecting advances in glaucoma and retinal shifts in diabetes. We thus identified 1762 applications of artificial intelligence in ophthalmology: review articles and research articles (301 pub med, 144 scopus, 445 web of science, 872 science direct). Of these, we analyzed 70 articles and review papers (diabetic retinopathy (N = 24), glaucoma (N = 24), DMLV (N = 15), other pathologies (N = 7)) after applying the inclusion and exclusion criteria. Conclusion: In medicine, AI tools are used in surgery, radiology, gynecology, oncology, etc., in making a diagnosis, predicting the evolution of a disease, and assessing the prognosis in patients with oncological pathologies. In ophthalmology, AI potentially increases the patient’s access to screening/clinical diagnosis and decreases healthcare costs, mainly when there is a high risk of disease or communities face financial shortages. AI/DL (deep learning) algorithms using both OCT and FO images will change image analysis techniques and methodologies. Optimizing these (combined) technologies will accelerate progress in this area.

## 1. Introduction

### 1.1. General Aspects

Artificial intelligence (AI) is acknowledged as a dramatic technological development, mainly due to the variety of applications in which its techniques can be used.

The tools of artificial intelligence have proved their effectiveness in modelling and optimization, with the most used being artificial neural networks (ANNs), evolutionary algorithms, and fuzzy systems. Their medical application in ophthalmology is a promising approach based on estimations and could enhance clinical examinations [1,2].

An artificial neural network is a simplified model of the biological brain. In its most general outline, a neural network is a machine designed to shape how the brain solves a particular problem or performs a function for a specific purpose (Figure 1).

In neural networks, information is no longer stored in well-defined areas, as in the case of standard computers, but is stored diffusely throughout the network. Memory storing is done by assigning appropriate values of the weights of synaptic connections between network neurons.

An artificial neural network takes over the biological characteristics of the biological network:-Information is stored and processed throughout the network—it is global, not local.-A key feature is the plasticity—the ability to adapt, to learn-Knowledge is stored in inter-neural connections (synaptic weights).-The ability to generalize—artificial neural networks can find the correct answers for slightly different inputs than those for which they were initially trained.-The ability to synthesize—can give correct answers for input affected by noise/inaccurate/partial.

Neural networks require processing units called neurons and channels through which information flows—interconnections. They are distinguished by a wide range of parameters (weights) based on the top neural connection. Each processing neuron first accounts for the weighted sum of all interconnection signals in a previous layer, to which it adds a basic term, resulting in an output via an activation function. The most employed networks consist of the following categories of layers: input, hidden, and output layer. The input layer is defined by the network input data, the output layer by the output data, and the topology of a network is outlined by the hidden layers (set of intermediate layers and of neurons within these layers) [1,3].

By adjusting the parameters in the connections between neurons, the network becomes able to learn from a set of numerical data suitable to the desired input and output variables. Therefore, based on wide range of examples, a key feature of RNA is that it can synthesize a specific pattern of the analyzed issue It is obvious that an essential advantage of neural networks is that they do not call for identifying a problem-solving algorithm, so it is not necessary to know the relevant laws, because the network alone learns from examples. Modelling based on neural networks consists of rendering the subordination between the output and input variables [1,2,4,5,6]. 

An essential feature of neural networks is their ability to generalize; after “learning” the behavior of the process, they can make projections regarding the “unseen” data (that were excluded from the training set), which defines their ability to generalize. It is the verification stage of the model represented by the network [7].

The neural network training is performed on a specific structure. The training process consists of establishing the network configuration, including architecture and topology.

Several methods are known for determining the topology of a neural network, of which the one based on successive tests is the most widely used. In this case, different topologies are tested until an acceptable error level is reached. This method, however, does not guarantee that the best network topology has been found. That is why evolutionary methods that have the chance to generate a favorable neural network are considered more efficient [8].

Multilayer perceptron (MLP) is the most acknowledged and extensively employed type of neural network. Most often, its processing units are interconnected in a feedforward way, meaning that the interconnections are not disposed in loops.

The advantage of using other artificial intelligence tools, such as machine learning, deep learning, and convulsional neural networks, is shown in the following sub-chapters, as they are useful through image analysis in the early diagnosis of some diseases. The field of machine learning has expanded to include deep learning and advanced neural networks such as the convolutional neural network (CNN). This is an advanced network, a multi-layered variant of DL, which simulates the interconnection of neurons of the human brain to analyze an input image (recognition, processing, and classification). Le-NET, AlexNet, VGG, GoogLeNet, and ResNet are some of the CNN algorithms.

### 1.2. ANN Ophthalmology Reviews

Having several medical applications, artificial intelligence (AI) tools started being employed in ophthalmology, such as detecting visual function deficits, thus playing a key role in diagnosing eye diseases and in predicting the evolution of these common and disabling diseases. Although relatively young in the field of ophthalmology, AI technologies are constantly expanding and have a significant impact on the scientific research and the improvement of clinical practice.

Artificial intelligence tools, and especially artificial neural networks, are progressively involved in detecting and customized control of ophthalmic diseases. The precise data of the explorations and, particularly, the emergence of new imaging methods (OCT) have led to the increase in interest in the use of these tools by multidisciplinary teams. The combination of AI technologies and optical coherence tomography (OCT) proved to be trustworthy in detecting retinopathies or in enhancing the diagnostic conduct of retinal diseases.

As in medicine, there are two types of approaches based on neural networks in ophthalmology: setting the database by image processing or using data from medical records. The first case is more common due to the possibility of using a lot of information from the image. Consistent databases can be both quantitatively and qualitatively generated. The second option is less used, precisely due to the difficulties in achieving a database suitable for processing.

There are a number of reviews in the literature that attempt to present the use of artificial intelligence tools in ophthalmology. We consider it necessary to refer briefly to these in order to highlight and justify our own contribution in this field and the originality of the approach.

In a short review [9], Grewal S., P., et al.’s *Deep learning in ophthalmology: a review*, deep learning tools were implemented for various diseases (cataracts, glaucoma, age-related macular degeneration, diabetic retinopathy) based on several diagnostic investigations (digital photographs, optical coherence tomography, and visual fields). Deep learning (DL) refers to machine-learning methods that use winding neural networks (CNNs), which employ storing image-processing filters to remove different types of image characteristics. The relevance and convenience of DL research in ophthalmology as daily clinical practice as part of digital ophthalmologic diagnostic tools is pointed out here. The approach reviews the benefits of DL techniques as a secure tool to render ocular data acquired from digital photographs and visual fields, and its contribution to early diagnosis of diseases such as AMD, DR, and glaucoma. Apart from its advantages for both patient and clinician, DL has also limitations related to modern technology and its best applications.

The mini-review of Kenji Karako et al. [10] describes applications of neural networks in medicine. However, as it addresses the classification of data types necessary for the neural networks training, this is also of interest for the field of ophthalmology. The first general consideration is that medical neural network applications can be divided into two types: automated diagnosis and physician aids.

Neural networks are being trained by employing different medical images to change diagnosis by a physician because diagnosis is often based on imaging, and a winding neural network activates image analysis. The network has the capacity to diagnose disease more accurately and even faster than a physician. Physicians can only identify a restricted amount of patients, while a neural network automated diagnosis can detect a substantial number of patients without time limitations.

The second type of applications refers to the use of data from medical records for neural network training, aiming to support the physician by generating a diagnostic rule.

A consistent and well-structured review is one by Lu W. et al. [11]—*Applications of Artificial Intelligence in Ophthalmology: General Overview*. The ophthalmological approach starts with the idea that the quantity of the image data is quickly increasing; therefore, investigating and changing these efficiently becomes a priority. AI methods can do this task very well, attempting to examine medical data and leading to remarkable development in establishing a diagnosis. Useful algorithms from the category of CML (Conventional Machine Learning), such as Decision trees, Random Forest, Support vector machines, Bayesian classifiers, k-nearest neighbors, k-means, Linear discriminant analysis, as well as CNN (Convolutional Neural Networks), are reviewed.

Various ophthalmic imaging methods in AI diagnosis were mentioned as potential sources for generating databases: fundus images, optical coherence tomography, ocular ultrasound B-scan, slit-lamp image, and visual field.

The main diseases discussed in detail from this point of view—the processing of characteristic images with AI tools—are diabetic retinopathy, glaucoma, age-related macular degeneration, and cataract.

AI applications can greatly contribute in order to provide support to patients in remote areas by sharing expertise, knowledge and methods. Clinicians also receive valuable help in diagnosis.

A comprehensive review (*Artificial intelligence and deep learning in ophthalmology*) by Daniel Shu Wei Ting et al. [12] deals with deep learning (DL) applied in ophthalmology for image recognition, as applied to fundus photographs, optical coherence tomography, and visual fields, reaching a powerful classification performance in detecting diabetic retinopathy and retinopathy of prematurity, the glaucomatous disc, macular oedema, and age-related macular degeneration. DL in ocular imaging may be employed linked to telemedicine as a potential solution to screening, diagnosing, and monitoring major eye diseases for patients in primary care and community environments. Considering the approaches of different researchers, the contribution of DL is explained in detail for diseases such as diabetic retinopathy, age-related macular degeneration, DM, choroidal neovascularization, glaucoma, and retinopathy of prematurity. The review of the 72 bibliographic references ends with highlighting and discussing potential challenges.

A short article (*AI papers in ophthalmology made simple*) by Sohee Jeon [13] points out, without many examples, the essential considerations for AI (particularly deep learning, DL) applications in ophthalmology. Three directions are briefly described: 1. Depending on the specific research question AI input may comprise clinical data, medical images, or genomics. 2. In AI processing, the most familiar type of DL method applied for medical images is winding neural network (CNN). 3. For AI output, the AI efficiency is measured by comparing it with a reference standard, which is often an extensively endorsed gold standard or ground truth. Having a key role in validating an algorithm, the reference standard is often grounded in compliance with several professionals, consultant ophthalmologists, fellowship-trained subspecialists, certified nonmedical professional graders, or optometrists who have engaged in extensive training and accreditation with restored and substantial outcomes. The results section quantifies the development of the AI system by relating to separating procedures, such as the area under the curve (AUC), sensitivity (or the genuine positive rate or withdrawal), peculiarity (corresponding to 1—false positive rate), positive predictive value (PPV), and negative predictive value (NPV).

Another review (*Artificial Intelligence: The Big Questions, Review of ophthalmology*, 2021) by Christine Leonard [14] presents the general problems of AI and the latest innovations and challenges, with some examples in ophthalmology, including detecting diabetic retinopathy, devices for diabetic eye-disease screening, cataract surgical videos (an AI algorithm to be able to automatically detect what steps are being performed in a surgical video at any given moment), predicting macular thickness from fundus photos, etc.

A short review (*Artificial intelligence and deep learning in ophthalmology—present and future*) by Moraru Angrea Dana et al. [15] explains Artificial Intelligence terms and focuses especially on deep learning and using OCT images for generating databases with the purpose of diagnosing DMO (diabetic macular oedema) and AMD (age-related macular degeneration) while using various algorithms such as (MLC, SVM, MLP, RBFNN). Other diseases, such as Diabetic Retinopathy and Retinopathy of prematurity, where deep learning was used are also mentioned. The review also focuses on limitations and future prospects, highlighting the importance of these technologies and how useful they can be, provided we can manage to standardize databases.

Different surveys in the literature review show the remarkable benefit of these AI tools in ophthalmology in evaluating the visual field, optic nerve, and retinal nerve fiber layer, thus ensuring a higher precision in detecting advances in glaucoma and retinal shifts in diabetes. All the studies that refer to these medicinal and opthalmological AI applications were analyzed in this review.

## 2. Methodology

The current review uses the standard procedures previously reported by Green et al. [16].

### 2.1. Database Searches

Between 2015 and 2022, all surveys were acquired by exploring the following data bases: PubMed/Medline, ISI Web of Knowledge, ScienceDirect, and Scopus. The research strategy comprised keywords to search for corresponding resources: “artificial intelligence”, “artificial intelligence in medicine”, “glaucoma and artificial intelligence”, “diabetic retinopathy and artificial intelligence”, “deep learning in ophthalmology”, “macular degeneration and artificial intelligence”, “applications of artificial intelligence in ophthalmology”, “artificial intelligence in obstetrics and gynecology research and clinical practice”.

### 2.2. Eligibility Criteria

English surveys published in or after 2015 were selected. Non-English articles, editorial letters, conference posters, preprints, and computational simulations were excluded.

### 2.3. Study Selection

Four independent authors (N.A., B.D., S.C., F.T.) checked the titles and abstracts of the recovered works; the ones that met the eligibility criteria were further examined for a full-text review. Any conflict was resolved by a third reviewer (C.M.B., C.L., and O.-D.I.). Accordingly, we used the best-justified accounts to detect the most relevant results.

## 3. Results

We identified 1762 results for artificial intelligence applications in ophthalmology: reviews and research articles (301 pub med, 144 scopus, 445 web of science, 872 science direct). Of these, we analyzed 70 after applying the inclusion and exclusion criteria. A review of applications in medicine shows the fields of application and types of neural networks used, further amplifying the importance of medical artificial intelligence (Table 1). Figure 2 shows a flowchart of the current survey design, strategy, results, and studies that complied with the eligibility criteria.

### 3.1. Brief Presentation of the Use of Artificial Neural Networks in Medicine

AI tools are known for their efficiency in modeling and optimization, with the most used being ANNs, evolutionary algorithms, and fuzzy systems. Neural networks are real modeling tools that have the ability to learn various types of relationships. Being able to approximate any continuous nonlinear function, neural networks offer an effective solution in modeling complex nonlinear systems. An ANN is a simplified model of the biological brain. In its most general type, a neural network is a machine built up to substantiate how the brain solves a specific problem or performs a function for a specific purpose. Networking is usually implemented using digital computer software [2]. In neural networks, information is no longer stored in well-defined areas, as in the case of standard computers, but is stored diffusely throughout the network. A relevant feature of RNA is that, based on several examples, it can synthesize a certain model of the issue [1,2,3,5]. An essential advantage of neural networks is that they do not call on the identification of a problem-solving algorithm; therefore, it is not necessary to know the relevant laws, because the network learns itself from examples. Modeling based on neural networks consists of rendering the dependence between the output and input variables [3,5]. An essential characteristic of neural networks is their potential to generalize; after they have “learned” the behavior of the process, they can make projections regarding the “unseen” data (that were excluded from the training set), which defines their ability to generalize. This is the verification stage of the model represented by the network.

Since the 1950s, researchers have been exploring potential applications of AI techniques in every field of medicine [17]. Among the first applications in surgery was that of Gunn who attempted to detect acute abdominal pain by computer scanning [18] in 1976. Stamey A.B. acquired a neural network-derived classification algorithm called the ProstAsure Index, which may differentiate prostate nodules into benign or malignant, a model acknowledged in forthcoming studies with a 90% accuracy, 81% sensitivity, and 92% specificity for the diagnosis [19]. Similar issues for the application of RNA in surgery refer to diagnosing acute appendicitis and residual gallstones [20,21]. The first commercially promoted ANN model was a computerized automatic screening system developed to assist the cytologist in screening for cervical cancer, called PAPNET. This model was later extended to the diagnosis of gastric [22] and thyroid [23] lesions, in differentiating oral epithelial cells [24], in identifying malignant urothelial cells [25], and in classifying the cells in pleural and peritoneal exudate [26]. Neural networks have been employed to render simple radiographs [27], ultrasound [28], computed tomography (CT) [29], magnetic resonance imaging (MRI) [30], and radioisotope scans [31]. Due to the ability of ANNs to recognize patterns, they have been used to analyze different waveforms: electrocardiography (EKG) interpretation to diagnose myocardial infarction or fibrillation [32], ventricular arrhythmias [33], electroencephalography (EEG) analysis in the diagnosis of epilepsy [34] and sleep disorders [35], analysis of electromyography waves (EMG) [36], and Doppler ultrasound [37]. Another application of neural networks is in establishing diagnosis based on computerized images; convolutional neural networks (CNNs) that facilitate image analysis were used, thus achieving a classification of lesions. This network includes an input, hidden, and an output layer. The images to be classified are inserted in the input layer. Based on the images in the hidden layers, the output layer categorizes images [10,38]. A relevant example is the Camelyon dataset (16)15, which uses a neural network to explore a series of pathological images from breast cancer patients; the neuronal model had a 92.4% accuracy in diagnosis, much higher than that of the pathologists (73.2%) [39]. Chest radiographs can be interpreted using a CNN. This approach was superior in the diagnosis of pneumonia compared to that of professional radiologists. Furthermore, the method identified 14 types of pathologies from a database of chest radiographs, with a higher success rate than other methods. Methods have been proposed to classify chest diseases using a trained neural network based on X-ray images [40,41]. Neural networks were even used [42] in the detection and classification of common dental caries. Likewise, neural networks have been employed in the early recognition of Alzheimer’s disease (AD) or in differentiating it from vascular dementia of the brain through single-photon emission computed tomography (SPECT) [43].

A neural network involved with CT, MRI, and positron emission tomography (PET) imaging has been proposed for the detection of brain tumors [44]. Another recent use of these networks is in predicting medical events and assessing the prognosis. It is also used in electronic medical records by designing a recurring neural network to estimate events such as symptoms, drug information, and visit schedules. The method used the recall function in a proportion of 79.58%, with 85% of the data being for training and 15% for the testing phase [45]. An extremely accurate method of predicting the medications to be taken by patients has also been proposed [46], using a recurrent neural network driven by data from electronic medical records. The assessment of the prognosis is highly significant in planning adequate treatment and follow-up strategies, which can help to cure or can prolong survival. It has been shown that neural networks can estimate survival in patients suffering from breast, colorectal [47,48], lung [49], and prostate cancer [50] better than those in the field. An interesting approach to AI is its use in obstetric gynecology. A recent review by Pulwasha Iftikhar et al. revises the relevant aspects of AI in obstetrics and gynecology (OB/GYN) and how it can be used to enhance patient outcomes and diminish healthcare prices and workload for clinicians. Thus, the use of AI as a tool for fetal heart rate interpretation (FHR) and cardiotocography (CTG) is used to help detect premature labor and pregnancy issues and evaluate inconsistencies regarding examination between physicians to lower maternal and fetal morbidity. The extensive storage output of AI data can help determine the risk factors for the use of premature labor [51]. An additional detail of IVF and AI is the capacity of determining the most valid oocytes and embryos. An AI system used by Manna et al. recommended blending AI to remove consistency descriptors from an image (local binary model) and grouping these by using an ANN. The suggested system was applied to two data sets of 269 oocytes and 269 corresponding embryos from 104 women and compared with other machine learning, methods already advanced in the past for close rating problems. Notwithstanding that the results are exploratory, they reveal a significant rating output [52].

The study that uses data and AI to generate a computerized algorithm that can predict pregnancy using in vitro fertilization (IVF) is interesting. Precise and fast prediction of the outcome of in vitro fertilization (IVF) treatment is relevant for both patients and physicians. Guh et al. introduce a hybrid method of AI that covers the genetic algorithm and decision learning techniques for extracting the information of an IVF medical database. The advanced method can not only help the IVF doctor in speculating the IVF outcome, but can also find available information that can help the IVF doctor to adapt the IVF treatment for each patient, according to the individual characteristics of the patient, in order to develop the highest chance of pregnancy [53]. Table 1 shows an overview of the major medical artificial intelligence applications. In conclusion, artificial intelligence finds its place in many medical applications. In gynecology, it allows the early prediction of the outcome of an IVF treatment, which is vital for both patients and doctors. The analysis of the images retrieved by X-ray, CT, and MRI, with the help of convolutional neural networks (CNN), allows classification of lesions.

**Table 1 diagnostics-13-00100-t001:** Medical artificial intelligence applications.

Domain	Input Data	Output Information	Neural Network Use	References
Surgery	Pathological images	distinction of prostate nodules as benign or malignant	RNA ProstAsure Index	[18,19]
Oncology	Pathological images	diagnosis of cervical lesions, gastric (13), thyroid (14) lesions in determining the oral epithelial cells (15), and identifying the malignant urothelial cells (16), as well as in classifying the cells in pleural and peritoneal exudate (17)	System PAPNET	[22,23,24,25,26,27]
	CT, MRI, radioisotopic scans	detection of brain tumors	RNA	[29,30,31,44]
Cardiology, neurology	EKG, EEG, EMG	diagnose myocardial infarction or fibrillation ventricular arrhythmias, EEG analysis in diagnosing epilepsy (25), and sleep disorder analysis of EMG (27) or Doppler ultrasound (28).	RNA	[32,33,34,35,36,37]
Cancer diagnosis, Pneumology diagnosis, dentistry	Pathology images, X-ray images	classification of images classification of breast cancer detection of pneumonia classification of chest pathologies classification of dental caries classification of X-rays reading chest X-rays identification of the spine and pelvis in frontal X-rays	convolutional neural networks (CNN)	[10,28,38,39,40,41,42,43]
Medical diagnosis		prediction of medical events and evaluation of the prognosis	ANN	[44,45,46,47,48,49,50]
Obstetrics and gynecology		as a tool for FHR and CTG; the possibility of determining the most valid oocytes and embryos for pregnancy prediction using IVF	ANN, genetic algorithm	[51,52,53]

### 3.2. The Use of Artificial Intelligence in Ophthalmology

In ophthalmology, artificial intelligence has several applications, with the most recent studies and common conditions for which artificial intelligence tools used in diagnosis and prediction being presented in this review: diabetic retinopathy, glaucoma, retinopathy of prematurity, DMLV, as well as other pathologies (cataract, neuro-ophthalmology, etc.).

#### 3.2.1. Use of Neural Networks in Diabetic Retinopathy

The increasing incidence of DR, with a growing spread of obesity and an aging population, has become a public health issue. Diabetic retinopathy (DR) is a particular microvascular issue of DM, and it affects one in three persons with DM. It is an eye complication that occurs over time, and it is linked with a low blood sugar control and increased blood pressure and blood lipids. According to the ETDRS, there are criteria for the classification of diabetic retinopathy due to the clinical changes of the fundus: no shifts (absence of DR), mild type of non-proliferative diabetic retinopathy (a single micro aneurysm), moderate type (micro aneurysms, hemorrhages in two to three quadrants, venous dilations and soft exudates in a quadrant), severe type (micro aneurysms, hemorrhages in all quadrants, venous dilatation in two to three quadrants) and proliferative diabetic retinopathy (disc and retinal neovascularization in different quadrants). Under these conditions, early undiagnosed DR is one of the leading sources of vision loss. ANNs have been used to analyze fundus images so as to spot early pathological changes. Thus, there are several studies currently analyzing the advance of software programs that improve the performance of DR screening based on different techniques: machine learning (MLC), machines with support vectors or multilayer perceptron neural networks or with basic radial functions that can recognize and classify DR in images [54,55,56,57]. Several surveys, such as those by Gulshan et al. [58] and Ting et al. [59], have used large databases to create MLCs. The neural model was able to identify hemorrhages, exudates, microaneurysms, and cotton and neovase points and was also capable establish a range of DR evolutionary stages. Other authors used the whole deep learning algorithm for the automatic detection of DR, processing color images of the fundus, classified as without retinopathy and with retinopathy at any stage. A total of 75,137 public images from Eye-PACS LLC, Berkeley, CA, were used to stimulate and examine an AI model to differentiate normal from pathological images. The results of the algorithm demonstrated the efficient potential of this program, since its implementation drastically reduced the rate of blindness attributed to DR [60]. Gardner et al. took eye photos from 147 diabetics (by analyzing the structure of blood vessels, hemorrhages, and exudates) and 32 normal images, projecting a back propagation neural network (the error signal is enhanced backward, from the output to the input layer). Comparing the RNA results with those achieved by the ophthalmologists, RNA had a 88.4% sensitivity and 83.5% specificity for the DR detection, due to its accuracy, with the system being used in the early detection of DR [61]. Wong et al. set the various stages of DR (moderate, severe, and proliferative) and altered from the healthy retina using three-layer neural networks and the backpropagation algorithm (BPA) for training. They focused on 124 retinal photographs (95 patients and 29 normal subjects) and analyzed the characteristic changes in retinopathy compared to normal images, attaining a sensitivity of 90% and specificity of 100% [62]. Garcia proposed a multilayer perceptron RNA (MLP), with radial base function (RBF) and a machine-based vector with support vectors (SVM) for DR exudate research [63]. A database that included 117 color images was used, with variable brightness and quality, from patients with DR and, as a reference, from healthy subjects, with the results being promising and achieving 100% specificity. Jen Hong Tan et al. developed an algorithm that uses 10-layer winding RNA to differentiate exudates, hemorrhages, and micro-aneurysms from DR [64]. The classification algorithms take into account all possible tests to select how to extract the most relevant information. In some studies, RNA has been used to identify pathological changes in OCT in patients with diabetes [65,66,67]. Another study (Wu et al.) that analyzed 240 images, including 120 eyes of patients with early DR and 120 normal eyes, developed a back-propagation RNA [68]. The improved BP-RNA network was compared to a traditional BP plus SVM and SVM network. The results showed that BP-RNA needed a shorter training time. The results were compared with those of previous papers [69,70,71]. The experimental method applied by the authors (Wu, 2020) includes previous knowledge that allows the extraction of features resulting from the processing of available images, which are also employed for assemblying the traditional neural network. Jonathan Krause et al. used the five-degree DR severity assessment scale: no retinopathy, mild, moderate, severe, and proliferative. DR gradation is an intricate process that demands recognizing and quantifying aspects such as microaneurysms, intraretinal hemorrhages, microvascular abnormalities, and neovascularization. Images from the EyePACS database were retrieved (between May and October 2015). The clinical set-up verification was performed by three retinal specialists and three general ophthalmologists. Based on these data, the deep learning algorithm for the prediction of DR and diabetic macular edema (DME) was created on the structure employed by Gulshan et al., represented by a CNN. The authors concluded that an appropriate predetermination as close to the truth as possible is useful in the early diagnosis of retinal diseases [72].

Also by using AI instruments, SVM combined with a strong optimization algorithm—differential evolution (DE)—we evaluated the shifts connected to DR (no shifts, small or moderate shifts) in patients suffering from glaucoma and diabetes in our recent study. In order to rate the DR shifts and to use various prognoses, an approximation including SVM optimized with DE was applied. The outcomes were relevant: a 95.23% accuracy was reached during the testing phase, with just one sample being incorrectly rated. These outcomes are similar to those revealed by the literature or even better [73]. In a recent study conducted with our contributors, we evaluated the shifts related to DR in patients with glaucoma and diabetes using AI tools: SVM combined with a strong optimization algorithm—DE. The combination of DE and SVM proved to be effective, with the methodology offering relevant results for the current issue: an accuracy of 100% for the training set and 95.23% for the test set, with only one sample being incorrectly rated. These outcomes are similar to those revealed by the literature or even better, because the DE algorithm leads to an optimized SVM model [73].

Various recent reviews reiterate the idea of using artificial intelligence in diabetic retinopathy screening. These suggest that AI systems that detect more than mild diabetic retinopathy and diabetic macular edema approved for use by the US Food and Drug Administration (FDA) are a substitute for long-established screening approaches. This is done by using retinal images to investigate the production of AI grading systems for DR detection. In this respect, AI systems have been detected to reduce costs, upgrade diagnostic precision, and enhance patient admission to DR screening [74,75,76]. The most recent review by Nakayama et al. reported in Table 2, describes open-access ophthalmic public datasets of fundus photographs, eyePACS, ODIR (Chinese public dataset), APTOS (public Indian dataset), DR1 and 2 (Brazilian public datasets), IDRiD (Indian public dataset), Jichi (Japanese public dataset), Rotterdam Ophthalmic Data Repository (ROD REP), Pathologic Myopia Challenge (PALM), and Tsukazaki, a public open-access Japanese dataset. They use several sets of photos acquired through high-performance digital systems, with a total of 131,459 collected images, showing around 0.01% of the global population. The public data sets are samples from the USA, China, India, Brazil, Japan, The Netherlands, and France, with 188 countries not depicted. ICDR (International Clinical Diabetic Retinopathy) was the most employed rating (five datasets with 104,556 images; accounting for 79.53% of the total). The authors consider that the use of artificial intelligence algorithms is not an impediment to use in developed countries; the problem arises in countries with low incomes when the purchase of the equipment to be used is an issue. AI models must also be reproducible, allowing transfer knowledge, implementation, and cross-validation. According to the authors, in spite of the technological advances, especially from a technical point of view, several questions prevent AI implementation globally [77].

#### 3.2.2. Use of Neural Networks in Glaucoma

Where there are doubts about the diagnosis, the most frequent employment of neural networks in ophthalmology occurs in the unexpected diagnosis of glaucoma. Neural networks have a major role in establishing the demand to initiate preliminary anti-glaucoma therapy to stop the advance of the disease.

The opening article that refers to the use of ANNs is that of Anton et al., which includes the use of ANNs in clarifying the newly developed perimeter lesions caused by glaucoma. The authors established that neural networks may discriminate, with a 97% accuracy, primary glaucoma lesions from those caused by other diseases [78]. In 2002, Bowd et al. used various AI tools to determine the evolutionary changes of the visual field in glaucoma patients and to predict the stage of glaucoma [79]. In 2005, the same author and contributors used two algorithms (RVM and SVM) to classify healthy eyes and glaucoma-affected eyes, using information based on the retinal nerve fiber layer (RNFL) and thickness measurements achieved by scanning laser polarimetry (SLP) [80]. Other authors have also used various AI tools (neural networks and learning algorithms) to induce a potential visual field advance in glaucoma patients [81,82,83,84]. Other AI tools, including SVM in combination with DE, have been developed since 1992 and used in ophthalmology only after 2010. Bernardes et al. [66], Girard et al. [85], and Jalan S. et al. [67], used them to identify the imaging changes on OCT in diabetic patients. The class algorithm SVM was used by Zheng et al. [86] to explore if diabetes and DR may alter the efficiency of Heidelberg Retina Tomograph II (HRT II Heidelberg Engineering, Heidelberg, Germany) in order to highlight the presence of glaucoma. LAS has been used to establish nonlinear relationships based on the principle of minimizing structural risk. The basic goal in LAS is that training courts are considered as points in a multidimensional space that can be changed so that classes become separable by a wide limit (Butnariu et al. 2013) [87]. The second algorithm used, DE (differential evolution) is influenced by the Darwinian evolutionary principle, belonging to the class of evolutionary algorithms. A multitude of studies using new algorithms manage to identify various types of glaucoma [88]. AI output has recently improved. In recent years, one of the deep learning algorithms, the recurrent neural network (RNN), has demonstrated remarkable skill in succession labeling and projection tasks for sequential data. The RNN model achieved by Keunheung Park et al. predicted a significantly better future visual field than a conventional point-to-point linear regression method. In clinical practice, the RNN model can help make decisions for the subsequent treatment of glaucoma. This RNN model has also been more robust in terms of reducing the reliability of visual input data [89]. After selecting the factors associated with open-angle glaucoma, through Multivariate Logistic Regression, these data are used to create several models for predicting the risk of glaucoma. Those factors that were beneficial in discriminating open-angle glaucoma from suspected glaucoma were sex, age, menopause, hypertension length, SERE, IOP, vertical cup-disc ratio, and superotemporal and inferotemporal RNFL defects. The authors use the ANN to calculate a risk calculation for glaucoma. Their conclusion is that ANN is a good screening tool for discriminating glaucoma patients from suspects and notably lowers the number of subjects who need a VF test to reinforce the confirmation of OAG. This becomes more accurate by adding new informative data thanks to ANN [90].

Recently, convolutional neural networks have developed and are used in image recognition [38,89,90,91,92,93,94], and recurrent networks employ sequential data for speech recognition [95]. The medical image unit is a significant element of medical image research. With the growing progression of convolutional neural networks in image processing, deep learning methods have achieved great success in medical image processing. Thus, most recent studies use optical disc imaging, developing an objective machine learning classification pattern for the classification of glaucomatous optical discs and revealing classification criteria to contribute to the clinical management of glaucoma [96,97,98]. Guangzhou et al. used images from 163 eyes, evaluated by glaucoma specialists; the image acquisition was performed by optical coherence tomography (OCT). A total of 91 parameters were selected that included basic information about the patients’ eyes. Neural automatic classification models were built using neural networks, Bayers networks, and SVM (support vector machine) and were trained to build classification models. The accuracy of the networks used was 87%; they were trained to classify glaucomatous optical discs with comparatively high efficiency without insisting on a color background image [88,89]. Other studies use the CNN (Convolution Neural Network) model (deep model learning), which uses the complete image of the optical disc, avoiding segmentation, to distinguish the normal structure from the structure of the optical glaucomatous disc. A 98% diagnostic accuracy was obtained using 1426 normal images and 837 glaucoma images. The model developed by the authors could be employed to reveal glaucoma at an early phase and to help with early counseling in the treatment of patients [94]. Multiple other recent studies using the fundus image and deep learning, as well as segmented optical disc images and optical cups, have proposed convulsive neural networks to increase the efficiency of the extraction module and different networks for optical disc and optical cup; they then calculate the disc cup ratio used as an indicator for glaucoma [95]. Yuming Jiang et al. use boxes to trace the limits of the optical disc and the optical cup, with the latter being inscribed as ellipses. They posit a convolutional neural network based on segmentation of optical discs and cups, called JointR CNN. The outcomes show small overlap errors on the optical disc and cup (6.3% to 20%), which are useful in the early detection of glaucoma [99]. By analyzing 1542 images achieved by photographing the fundus of the eye (red free photography of the RNFL), using the convulsive neural network (Deep Convolution Network and Resnet), another study conducted by Jin Mo Ahn et al. correctly detected both early and advanced glaucoma using only the bottom photos of the eyes, with an accuracy of 92.2% [100].

In another paper (2018), artificial neural networks were firstly employed to show the link between glaucoma and diabetes, as well as to anticipate the progression of ocular changes due to diabetes (diabetic retinopathy) in patients with glaucoma and diabetes. The constructed neural models demonstrated the possibility of their use in predicting MD (mean deviation) depreciation, with the best results being achieved by using JEN networks (Jordan Elman neuronal model) [3].

The most relevant studies use artificial intelligence in detecting glaucoma progression. Although a variety of methods are used to determine glaucomatous fundamental and useful shifts that occur over time, there is currently no accepted evidence of advanced changes. Using functional parameters from standard automated perimetry and fundamental changes on OCT (global and structural SNFR thickness) or using only the thickness of the retinal nerve fiber layers on the fundus image, learning algorithms that classify glaucomatous eyes and early detection of glaucoma were created [101,102]. Elsewhere, others have proven that the AI algorithm, Fusion Net, using VF data and circular peripapillary OCT scans in differentiating patients with glaucomatous optic neuropathy from those without this deficiency concluded an AUROC (area under the curve) of 0.950, outperforming the data provided by VF in comparison to OCT data and two glaucoma specialists. Thus, through machine learning techniques and through uncontrolled AI techniques, the structure-function connection and further prediction of VF loss rate was improved [101,103]. Other studies have added various data related to demographics, including information about intraocular pressure, visual acuity, and central thickness of the cornea, with the anterior surgical interventions being recorded in electronic health records (EHRs) and used as inputs in deep learning. The obtained models were able to predict subsequent glaucoma surgery [104]. Studies demonstrate that AI algorithms have the resources to include fundamental or useful changes over time and to ensure more precise and advisable recognition of glaucoma advances [105]. The images below schematically show the method of acquiring and selecting information. In another recent work (2022), various machine learning algorithms aiming at estimating the progression of open-angle glaucoma (POAG) were used. The rating of glaucoma advancement was established on grounding parameters such as VFI (Visual Field Index), MD (Mean Deviation), PSD (Pattern Standard Deviation), and RNFL (Retinal Nerve Fiber Layer). As rating tools, the following algorithms were used: Multilayer Perceptron, Random Forest, Random Tree, C4.5, k-Nearest Neighbors, Support Vector Machine, and Non-Nested Generalized Exemplars. The best results of over 90% accuracy were achieved by Multilayer Perceptron and Random Forest algorithms [106].

Another recent study was carried out on data from 73 eyes, with the role of constructing neural models to determine the advance of glaucoma in patients with sleep apnea. Therefore, 21 indexes were chosen as the input indexes for neural models, including patients’ age, BMI (body mass index), systolic and diastolic blood pressure, intraocular pressure, central corneal thickness, corneal biomechanical parameters (IOPcc, HC, CRF), AHI, desaturation index, nocturnal oxygen saturation, remaining AHI, and type of apnea, while the related extensive rating (diabetes, hypertension, obesity, COPD) selected output parameters were c/d ratio, modified visual field parameters (MD, PSD), and ganglion cell layer thickness. The findings reinforce the outcomes in other studies and strengthen the link between sleep apnea syndrome and glaucoma changes [107].

The conclusion of the glaucoma studies is that this innovative area should be viewed as a tool for clinicians who face the challenge of providing high quality glaucoma care outcomes for an aging population. In the future, AI may become a key support to glaucoma diagnosis; it will not replace cover clinical skills but will enable decision making [102,105]. Figure 3 shows the theoretical chart of the classification algorithm used in studies to predict glaucoma.

#### 3.2.3. Use of Neural Networks in AMD

AI has significantly contributed to the progress in screening for AMD (age-related macular degeneration), a common disease that occurs with an aging population, with disabling potential including irreversible loss of vision, and can help both patients and physicians make therapeutic decisions. Lee et al. created an AMD screening system capable of differentiating normal images from those with AMD on OCT. They achieved a sensitivity of 0.926, a specification of 0.937, and an AUROC of 0.9746 [108]. In a larger study, Treder et al. used 1112 OCT images to create MLC software that differentiates a pathology-free macula from one with exudative AMD lesions and achieved a sensitivity of 1.00 and a specificity of 0.92 [109]. Other AI-based studies attempted to solve problems such as if and when anti-VEGF therapy is required in the treatment of AMD. Prahs et al. [110] and Schlegl et al. [111] found that a deep learning network manages to correctly indicate the need for intravitreal injections in 95% of cases. The researchers looked at different features of OCT scanning, especially the presence of intraretinal and subretinal fluid. Central retinal thickness and fluid location are relevant biomarkers in OCT images. A deep learning algorithm was used to segment all baseline OCT scans [112,113,114,115].

In a recent article, Tiarnan D.L. Keenan et al. evaluated the efficiency of the Notal Vision Home OCT (NVHO), which used a spectral field OCT device for patient self-imaging at home, telemedicine infrastructure for automated data upload, and a deep learning algorithm for evaluating automatic OCT. The study aimed at analyzing the performance of the system in daily image acquisition and automated analysis and featured the dynamics of retinal fluid exudate in neovascular age-related macular degeneration (nAMD). The authors’ conclusion was that home-based OCT telemedicine systems are an alternative for disease monitoring and could enable highly personalized retreatment decisions, with fewer injections and clinic visits [116,117]. Li Dong, in a recent meta-analysis, quantified AI efficiency in exploring AMD on fundus photographs. After analyzing 13 studies according to the criteria of inclusion, a specificity and sensitivity of 0.983 indicated that AI could detect AMD in color fundus photographs. The AI-based automatic tools application is thus beneficial in diagnosing AMD [118].

#### 3.2.4. Use of Neural Networks in Retinopathy of Prematurity

Retinopathy of prematurity is a vasoproliferative disorder that concerns premature infants and can cause blindness. Treatment is based on the International Classification of ROP (ICROP) guidelines. Some deep learning algorithms may reduce the variability and inconsistency of the diagnosis of retinopathy of prematurity. Brown et al. [119] involved a deep convolutional RNA on a set of 5511 images, previously evaluated by experts. Validation of a set of 100 retinal images showed 93% sensitivity and 94% specificity for disease detection and sensitivity of 100% and 94% species in the detection of “pre-plus” disease. Corresponding studies have also shown that deep learning techniques may reduce the shift between observers [119,120].

#### 3.2.5. Use of Neural Networks in Cataract and Other Pathology

Cataract is one of the main sources of visual impairment worldwide. However, compared to other major age-related eye diseases such as diabetic retinopathy, age-related macular degeneration, and glaucoma, the advencement of AI in the field of cataract is still relatively underexplored. There is a number of recent surveys that use AI in detecting the cataract and its grading depend on the location and the severity of the changes. The images are obtained with a slit lamp and by backlit images to objectify the location of the changes. With the help of an AI platform based on deep learning DL, the precise detection and classification of cortical, nuclear changes was successfully achieved [121,122].

NO (nuclear opacities) and NC (nuclear-cortical opacities) in the seven-level classification and CO (cortical opacities) and PSC (posterior sub-capsular opacities) in the six-level classification exceeded medical limitations. A recent literature review summarizes the original articles on adults without restrictions on the study design. Of the initially extracted 1617 articles, 44 abstracts complied with the inclusion/exclusion criteria. With technological developments, more AI-based imaging tools and smartphone applications have emerged to ensure clinical decision assistance. Such an application can provide eye care through triage, diagnosis, and monitoring [123]. Another recent literature review shows current development in AI in the preoperative, intraoperative, and postoperative phases of cataract surgery, showing its effect on the evolution after surgery. The optimal intraocular lens (IOL) power to achieve the claimed postoperative refractive outcome can be counted with higher precision using AI-based modeling compared to traditional IOL formulations. Throughout the surgery, innovative and advanced AI-based footage analysis tools are developed, promoting a paradigm shift for the documentation, storage, and cataloging of surgical video libraries applied to teaching and training, complication review, and surgical research [124].

In comparison to other ophthalmic subjects, neuro-ophthalmology has not benefited from meaningful progress in artificial intelligence until recently. A recent review summarizes and discusses advances in using artificial intelligence to detect fundamental and severe abnormalities of the optic nerve head and eye movement disorders. Using machine learning, deep learning, and fundus photographs, an automatic system that detects different degrees of papillary edema (moderate vs. severe) compared to the healthy optic nerve and differentiates glaucomatous optic atrophy from optic neuropathies was developed. The Brain and Optic Nerve Study with Artificial Intelligence (BONSAI) showed high accuracy for classifying papilledema from normal subjects and other optic nerve abnormalities [125].

In ophthalmology, the most frequent areas of use of AI and classification tools were in diabetic retinopathy, glaucoma, and AMD. By using fundus images, OCT images, and visual fields, using classification algorithms, early diagnosis of lesions and prediction of their progression could be achieved. The advancement of AI in the field of cataracts is still relatively underexplored. The advantages of using AI in calculating the power of the lens implant compared to traditional formulas and the results obtained by residual refraction analysis are still being analyzed.

## 4. Discussion and Perspectives

This manuscript highlights the medical and ophtalmological applications of artificial intelligence in a wide range of ophthalmological diseases.

In medicine, artificial intelligence has been used to interpret simple radiographs [27], ultrasound, computed tomography (CT), magnetic resonance imaging (MRI), and radioisotope scans. Following the evaluation of the literature review, it is found that an increasing number of fields use artificial intelligence in the primary diagnosis of disease by creating classification algorithms that, based on the images, identify the difference between pathological and healthy.

An interesting approach in ophthalmology is applied in cataract surgery, where recent studies show a promising role in the precise calculation of IOL power as well as three-dimensional calculation in areas based on combining several existing formulas, thus producing a super-formula. Another application in cataract surgery would be a commercially available AI system for ophthalmic surgery. Touch Surgery Enterprise is a video management and analytics solution for surgeons in this respect. Additionally, these systems enable the examination of surgical tools and the tracking of resources and materials. The growth of AI-based technologies, especially in cataract surgery, makes this tool one that will be prominent in the near future [124]. Another successful use of artificial intelligence is in corneal pathology by differentiating between infectious and fungal keratitis, early detection of Fuchs endothelial dystrophy, as well as the detection and grading of keratoconus and predicting the results of placing intrastromal segmental rings. In the future, with the increased need for screening of corneal diseases and cataracts, both image-based and non-image-based AI algorithms could enable timely diagnosis and treatment of corneal diseases and advanced fields of cataract [126].

In the future, the medical application of neural networks can be focused in two directions: automatic diagnosis and help for doctors. Currently, 45% of WHO member countries count less than one doctor per 1000 inhabitants. Automatic diagnostic systems that use neural networks are in great demand for risk-free patient evaluation, thus preventing the overburdening of doctors and clearly establishing a rhythm of visits. There is a number of specialties in which automatic diagnosis is possible in imaging (X-rays, fluoroscopes, ultrasonography, CT, MRI) that address common, disabling diseases in the pathology of the elderly that pose public health issues: cardiovascular disease, cerebromedullary, and oncological diseases. They will drive the development of minimally invasive methods such as interventional radiology, interventional cardiology, and interventional neuroimaging [10].

Another future direction is prediction of a medical event that enables a doctor to know where to focus his or her attention in the moment. Based on the prediction of a patient’s next visit, the doctor may have the patient come to the hospital earlier so that the symptoms do not worsen. An example is the use of electronic medical records to estimate events such as symptoms, drug information, and visit schedules. The neural network was loaded with data from electronic medical records from 260,000 patients and 2128 physicians over a period of 8 years to specify the data and purpose of future medical examinations. The method used the recall function in a proportion of 79.58%, with 85% of the data being for training and 15% for the testing phase [127]. The assessment of the prognosis is also extremely relevant to plan appropriate treatment and follow-up strategies, which can help to cure patients or prolong survival. Neural networks have been shown to predict survival in patients with cancer (breast, colorectal [47,48], lung [49], and prostate [50] cancer better than those in the field.

One system that redefined the boundaries of science is the CRISPR-Cas9, due to its applicability in the medical field. Even though in the current literature, scarce data and relatively few studies aiming to renew the actual state of knowledge [128] can be found, the inclusion of CRISPR-Cas9 within AI algorithm(s) as an alternative therapeutic system has been recently developed. In this context, the proposed approach is in its infancy and needs to be optimized due to the mutations that occur, with these being reflected by a plethora of phenotypic changes. This is why a team of researchers synthesized a library that contains 41,630 pairs of distinct guide RNAs to target DNA sequences. Specifically, the team studied these constructs in a range of genetic possibilities and the ability of CRISPR-Cas9 reagents to analyze DNA cuts and repair processes. Altogether, they generated data for over 1 billion mutational outcomes, with the resulting program being called FORECasT, dedicated to predicting the status of repair, single-base insertion, and small deletions genes [129]. Another article using the same design was also published in the same year. The authors used a 2000 guide RNA-target pairs library and successfully developed a machine-learning model that was able to predict >50% of both insertions and deletions in patient-derived cell lines for three human disorders [130]. From that point onwards, most series of new manuscripts were published annually, explaining and bringing new evidence into the area of genome editing and how deep/machine-learning approaches have changed the course of therapy by significantly improving efficiency, using state-of-the-art models for avoiding unintentional off-targets (e.g., CRISPRLearner, CRISTA, C-RNNCrispr) [131,132,133,134]. These use deep learning/CNNs or hybrids or machine learning frameworks that are currently proposed. Based on these discoveries, the protocols that involve exon skipping [135], anti-CRISPR protein families [136], and base editors [137] have opened a new area of research.

## 5. The Advantages and Limitations of Using Artificial Intelligence Tools

The evolution of artificial intelligence has opened new paradigms in several fields of medicine. The most promising AI tools are currently in the retinal field—for diabetic retinopathy (DR), age-related macular degeneration (AMD), and retinopathy of prematurity (ROP). There are AI models applicable to glaucoma, keratoconus, cataracts, and other anterior segment diseases and oculoplastic surgery. Deep learning and convolutional neural networks (CNNs) extend the depth of layers with top prediction efficiency compared to traditional ML (machine learning) algorithms. A large amount of graphics processing units (GPU) memory and speed is required to implement CNN.

This type of network trained on small datasets would lead to the algorithm performing poorly (output) due to overfitting. The construction of data sets represents a difficulty for the data being a decisive stage in the development of machine learning algorithms, covering their representativeness to carry off correctness in AI applications.

Despite technological development, several challenges hinder the real-world AI implementation, such as variability in algorithm efficiency, patient confirmation of automated processes, and ethical conflicts. A key aspect is that of consent in applying these models. The use of artificial intelligence on a certain ethnic group may be challenging if an AI algorithm is trained or tested on a data set in which just a few groups of the population are represented. Another problem is of an ethical nature, because the implementation of artificial intelligence in a complex health system could lead, in complex cases with various other comorbidities, patients to lose their priority.

Another real problem is implementation in less developed countries due to the reduced financial resources in purchasing programs, informing patients about programs, and acquiring data (acquisition of images, standardized images, and their interpretation protocols); other issues include consensus regarding standard diagnostic criteria and signing the informed consent regarding data protection, confidentiality, and cyber security.

## 6. Conclusions

In medicine, AI tools are used in surgery, radiology, gynecology, oncology, etc., in making a diagnosis, predicting the evolution of a disease, and assessing the prognosis in patients with oncological pathologies. AI allows early prediction of the outcome of an IVF treatment, which is important for both patients and doctors. The analysis of images obtained on X-ray, CT, and MRI with the help of convolutional neural networks (CNN) allows a classification of lesions.

In ophthalmology, the most frequent areas of use of AI and classification tools are in diabetic retinopathy, glaucoma, and AMD. By using fundus images, OCT images, and visual fields, using classification algorithms, early diagnosis of lesions and prediction of their progression could be achieved. The advancement of AI in the field of cataracts is still relatively underexplored. The advantages of using AI in the calculation of the power of the lens implant compared to the traditional formulas and the results obtained by analyzing the residual refraction are still being analyzed. In ophthalmology, AI potentially increases the patient’s admission to screening/clinical diagnosis and decreases healthcare costs, mainly when there is a high risk of disease or communities face financial shortages. AI/DL (deep learning) algorithms using both OCT and FO images will change image analysis techniques and methodologies. Optimizing these (combined) technologies will accelerate progress in this area.

Currently, there are algorithms that regulate OCT images from different devices, and the outcomes of these software packages will be valid if values are substantially used.

AI is valuable in detecting different diseases, but ultimate recommendation will be made by the clinician.

The electronic health records could contribute to the development of this field and to reducing the time required for a clinician to establish an accurate and rapid diagnosis.

## Figures and Tables

**Figure 1 diagnostics-13-00100-f001:**
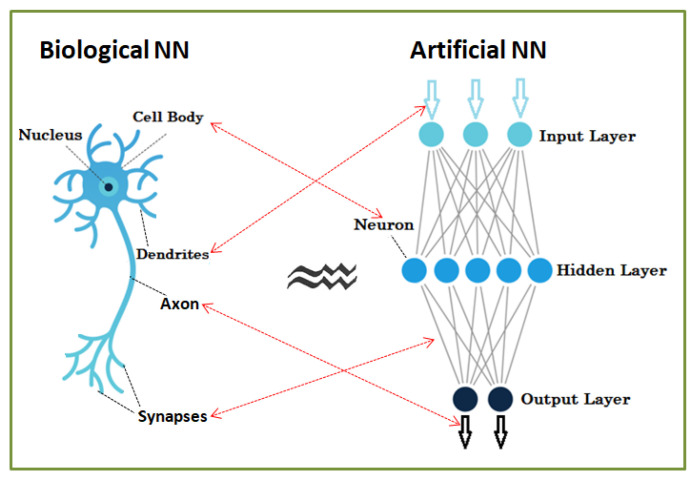
Neural networks—biological and artificial.

**Figure 2 diagnostics-13-00100-f002:**
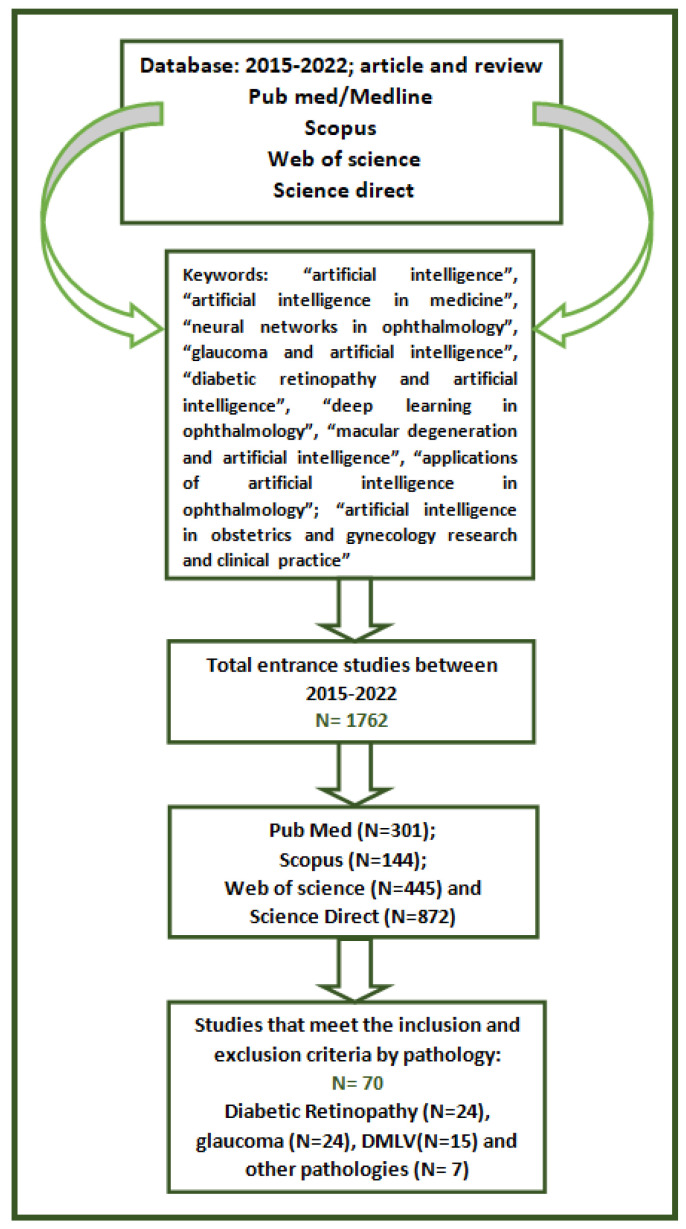
A flowchart of the current survey design, strategy, results, and studies that complied with the eligibility criteria.

**Figure 3 diagnostics-13-00100-f003:**
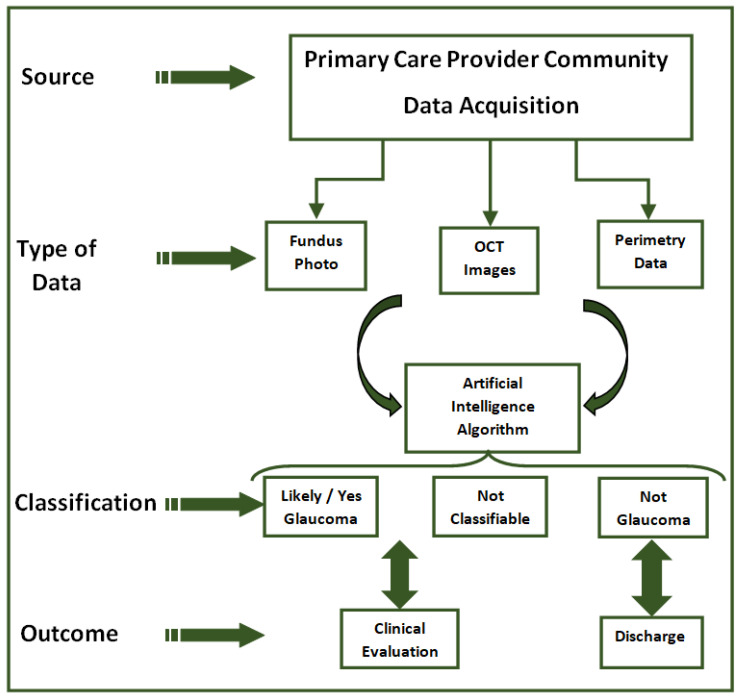
The theoretical chart of the classification algorithm used in predicting glaucoma.

**Table 2 diagnostics-13-00100-t002:** The open-access public ophthalmological datasets of retinal photographs used in a number of countries. International Clinical Diabetic Retinopathy (ICDR).

Public Datasets	EYEPACS	ODIR	APTOS	DR 1 and 2	IDRiD	Jichi	ROD Rep	Messidor 2	Tsukazaki	PALM
Images	88,702	8000	5590	1597	516	9939	1120	1748	13,047	1200
Country	USA	China	India	Brazil	India	Japan	Netherlands	France	Japan	China
Grading DR Glaucoma Cataract others	ICDR no	ICDR yes	ICDR	None	ICDR	Mod Davis	Not specified	ICDR	None	Not applicable
Sex, age, quality control, socio-economic aspects, or ethnicity	yes	yes	no	no	no	no	no	yes	yes	no

## Data Availability

The datasets used and analyzed during the current study are available from the corresponding author on reasonable request.

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
