# Peer review of "Comprehensive Review on the Use of Artificial Intelligence in Ophthalmology and Future Research Directions"

_diagnostics, 2022, doi:10.3390/diagnostics13010100_

Round 1
Reviewer 1 Report
1. Introduction has focused only on the very basics of the neural network rather than giving emphasis on Artificial intelligence (AI). Since the title focus on AI, more emphasis is required in the introduction.
2. Need to provide more citations for the introduction
3. Give a short discussion on ophthalmology disease conditions and the significance of the usage of ANN for such disease handing
4. Figure 1 need reorganization with the more professional touch
5. In line number 311 author quoted a number (3635). The paragraph does not clearly discuss this. Mention the details.
6. Sentence 651, 699 need modification ( typo need a clear check)
7. In the discussion section compare recent AI works carried out in the proposed
Author Response
Thank you for suggestions! The datailed point by point responses to your comments are given below! We do the changes!
Kind regards,
Nicoleta Anton

Reviewer 2 Report
This paper reviews using AI in medical diagnostics. They did a detailed survey of several applications and had good language writing. However, the following suggestions could be referred.
Many paragrams are very short, especially for section 1.2. Please reorganize them into better paper structures. In section 1, more AI applications may be added to extend the scope such as "Multi-needle detection in 3D ultrasound images using unsupervised order-graph regularized sparse dictionary learning" and "Multi-instance discriminative contrastive learning for brain image representation". In section 2, the authors could give a workflow that shows their survey works. In the results, I suggest authors have a conclusion for each ophthalmology showing the contributions of AI to the medical route. Please add more discussions on advantages and limitations from multiple views, e.g., data problems, subject problems, and route problems. Finally, the conclusion could be given a better presentation to show the contribution of this investigation and shortcomings while how to improve the survey for hospital routes.
Author Response
Thank you for suggestions! THe detailes point -by- point responses to your comments are given below! We do the Changes!
Kind regards,
Nicoleta Anton

Round 2
Reviewer 2 Report
I have no further comments.